## Research Article

mental health; risk perception; academic persistence; student retention; protective factors

**Corresponding author:**
Yusen Zhai;
Email: yzhai@uab.edu

# The role of mental health and protective factors in student academic persistence and retention during a global crisis

Yusen Zhai[1] and JoLynn V. Carney[2]

[1]Department of Human Studies, The University of Alabama at Birmingham, Birmingham, AL, USA and [2]Department of Educational Psychology, Counseling, and Special Education, The Pennsylvania State University, University Park, PA, USA

## Abstract

The COVID-19 pandemic has exacerbated challenges for millions of students globally, leading to enrollment cliff. This study addresses the existing research gap by investigating the influence of students' mental health and various protective factors (i.e., optimism, help-seeking behaviors, social support) on academic persistence, an indicator of student retention. We utilized the structural equation modeling approach to examine the effect of students' mental health conditions, risk perception of COVID-19 and protective factors on academic persistence through a sample of 1,051 students from 45 states. Students' mental health positively predicted academic persistence. Risk perception of COVID-19 was negatively associated with mental health but positively predicted academic persistence and help-seeking behaviors. Optimism fully mediated the effect of mental health on help-seeking behaviors but did not mediate the effect of risk perception on help-seeking behaviors. Social support positively predicted academic persistence. This study underscores the integral role of mental health and protective factors in supporting student retention. Universities should develop targeted programs to address students' mental health needs and promote protective behaviors. These initiatives can enhance academic persistence, thereby aiding in the retention of students affected by this pandemic or any future global crisis.

## Impact statement

As the world grapples with the consequences of the COVID-19 pandemic, the mental health toll on college students is substantial, and it has negatively affected their academic persistence and overall retention in higher education institutions. Our research sheds much-needed light on this crucial area, investigating the interplay between students' mental health, perceived COVID-19 risk, protective factors such as optimism, help-seeking behaviors and social support, and their impact on academic persistence. Our findings contribute to a deeper understanding of how these interconnected factors can shape educational outcomes during a pandemic or other crisis. By addressing these key issues, the study offers valuable insights to educators, administrators, policymakers and mental health professionals, enabling them to devise targeted intervention strategies that support student retention and mitigate educational disparities. The general public, particularly families with college-going students, can use these insights to better support their children's educational journey during such unprecedented times. Meanwhile, students themselves can gain a greater understanding of the protective factors that can bolster their resilience and academic persistence. In essence, our research speaks to the urgent need for comprehensive strategies that promote students' mental health and academic persistence in the face of global challenges – a contribution that has the potential to foster more resilient educational communities globally.

## Introduction

The COVID-19 pandemic has potentiated mental health concerns and amplified extant challenges among college students globally. During the academic year 2020–2021, an estimated 30% of U.S. college students confronted mental health problems (American College Health Association, 2021; Zhai and Du, 2022b). Concurrently, higher education institutions registered a significant enrollment decline, exacerbating the educational disparities among students, particularly those disproportionately disadvantaged by the pandemic-induced socioeconomic turbulence (Copley and Douthett, 2020). To redress these disparities, the development of targeted prevention and intervention strategies, focusing on student support and retention, is exigent. Nonetheless, a research gap exists regarding the factors that contribute positively to student

retention. This study endeavors to bridge this gap by examining the role of students' mental health and protective factors in retention in responding to the risk of COVID-19.

### Student retention and academic persistence

In higher education, student retention refers to the percentage of students who re-enroll semester after semester until they complete their degrees. Academic persistence, a critical determinant of student retention, is an individual student's sustained effort and behaviors to remain enrolled and actively engaged in their education despite challenges or distractions that might otherwise lead to withdrawal before completing their program (Tinto, 1975, 1993). Student retention is an institutional-level outcome that stems from the collective individual-level academic persistence efforts and behaviors of students. Considering our goal to assess students' responses to the pandemic, we employed academic persistence as a proxy to estimate individual-level student retention.

Tinto's theory of student departure suggests that academic persistence includes several key variables that influence student departure (i.e., dropout behaviors), such as students' academic and social integration, and their commitment to goals and the institution (Tinto, 1975, 1993). Previous research has shown that students' academic persistence results from a complex set of interactions between personal and environmental factors within the higher education context (Tinto, 1975, 1993; Cabrera et al., 1992; Aljohani, 2016). These interactions contribute to varying levels of students' integration into the academic and social frameworks of the institution, their attitudes toward the institution and their commitment to academic goals (Tinto, 1975, 1993; Cabrera et al., 1992). Measuring these pivotal variables of academic persistence can shed light on the dynamic and ongoing interactions between students and environmental factors. Such an evaluation can be critical in informing the development of institutional policies and strategies to improve student retention, particularly given the recent disruptions in higher education caused by the pandemic (Purcell and Lumbreras, 2021). These disruptions have significantly affected the academic and social structures of higher education, profoundly altering the interactions between students and higher education systems (Purcell and Lumbreras, 2021). It is anticipated that some students may have experienced diminished academic persistence, leading to worsened retention rates. However, it remains unclear why some students demonstrated stronger academic persistence than others. Therefore, a detailed examination and understanding of the factors contributing to academic persistence could have important implications for addressing student retention issues.

### Impact of mental health on academic persistence during the pandemic

Although previous studies have delved into the mental health issues faced by students during the initial stage of the pandemic (Cao et al., 2020; Wang et al., 2020), the influence of mental health on academic persistence – a critical determinant of student retention – under the threat of COVID-19 remains largely unexplored during this critical phase of the global crisis The pandemic has disrupted students' daily routines, learning experiences and social engagement, causing prevalent mental health problems (Zhai and Du, 2020; Purcell and Lumbreras, 2021). The severe morbidity and mortality associated with COVID-19 infections may lead students to perceive an elevated risk of COVID-19 to their health and safety, potentially intensifying stress and anxiety and exacerbating mental health issues (Finset et al., 2020). As individual factors can contribute to key variables of academic persistence (Tinto, 1975, 1993), the mental health conditions of students, coupled with their perceived risk of COVID-19, might significantly shape their academic persistence during the early phase of the pandemic. Nevertheless, empirical evidence delineating the effects of students' mental health and COVID-19 risk perception on their academic persistence remains scant. Therefore, the first aim of this study is to investigate the effects of students' mental health and risk perception of COVID-19 on academic persistence.

### Protective factors

Protective factors, including optimism and help-seeking behaviors, can contribute significantly to enhancing student retention by promoting academic persistence (Tinto, 1993, 2017). These factors enable students to withstand adversities and continue their educational pursuits in challenging scenarios such as during a pandemic (Skalski et al., 2020). Optimism has been associated with resilience in the face of adverse experiences (Uchida et al., 2018), while help-seeking behaviors have been identified as robust predictors of positive student outcomes, including academic achievement (Tinto, 1993; Li et al., 2014; Zhai et al., 2023). However, an optimism bias, defined as the belief that one is less susceptible to negative events compared to others, may paradoxically reduce students' help-seeking behaviors in response to perceived COVID-19 risk and attendant mental health issues (Park et al., 2021). To assess the role of optimism, our secondary aim is to investigate whether optimism would mediate the effects of students' mental health and risk perception of COVID-19 on their help-seeking behaviors.

Social support, another essential protective factor, functions as a buffer against distress (Steinhardt and Dolbier, 2008; Zhai and Du, 2022a). Students endowed with robust social support networks, and thus more inclined to seek help, are better poised to manage stress, potentially enhancing their likelihood to demonstrate academic persistence and degree completion during crises (Gloria and Ho, 2003). However, the reasons behind the absence of these protective factors in some students, rendering them more prone to discontinuing their education during the pandemic, remain ambiguous. Thus, the third aim of this study is to evaluate the extent to which optimism, help-seeking behaviors and social support mediate the effects of students' mental health and risk perception of COVID-19 on their academic persistence.

### Current study

College students' individual factors, such as mental health conditions, risk perception and protective factors, may contribute to the prediction of academic persistence variables, particularly during a time of public health crisis. However, there has not been an integration of these individual factors into contemporary models of student academic persistence (Tinto, 1975, 1993). This exploratory integration could help bridge the gaps in the literature concerning the potential role of mental health and protective factors in student academic persistence and retention. Grounded in Tinto's theory of student departure (1975, 1993, 2017) and the psychology of psychosocial factors, we constructed a hypothesized model to test paths that potentially illustrate the directional relationships between aforementioned variables. Specifically, our hypothesized model

suggests that students' mental health conditions and their perceived risk of COVID-19 correlate with each other and influence academic persistence directly and indirectly through mediators (i.e., optimism, help-seeking behaviors, social support).

These mediators are important protective psychosocial factors that directly or indirectly affect students' interactions with the higher education environment (Tinto, 1993, 2017). Together, they may be considered as one construct (i.e., protective factors) in the model; however, this approach fails to capture the intricate inter-relations between each protective factor and their associations with mental health, risk perception and academic persistence. Thus, we included these protective factors as three separate mediators rather than one higher-order construct to better understand their potential mediating roles within our conceptual framework. For example, optimism may mediate the effect of mental health issues and risk perception on help-seeking behaviors. Optimistic students are likely to perceive challenges, including mental health issues and the threat of COVID-19, as surmountable obstacles rather than insurmountable threats. This positive outlook can foster a greater inclination toward seeking help, as they are more likely to view such actions as effective steps toward problem-solving and coping (Carver and Scheier, 2014). One study (Aspinwall and Taylor, 1992) indicates that optimistic individuals exhibit more adaptive coping strategies, including seeking social support and professional help in the face of stress. Hirsch et al. (2007) found that optimism was linked to less avoidance-oriented coping and more approach-oriented coping, which is closely related to proactive help-seeking. Tinto (1993) emphasizes the importance of individual attributes in shaping students' engagement with their educational environment. In the context of the pandemic, optimism can be seen as a key individual attribute that influences how students perceive and respond to the compounded challenges of mental health issues and COVID-19 risk perception. Optimistic students, guided by a positive outlook, may be more likely to engage with available support systems, aligning with Tinto's perspective on the significance of individual agency in academic and social integration.

Further, research has revealed that help-seeking behaviors contribute to increased social support (Mishra, 2020). Engaging in help-seeking behavior often requires interaction with peers, family members and professionals. This interaction can lead to the formation of meaningful relationships and the expansion of social networks among college students. As postulated by Newman (2000), help-seeking is a social process that inherently involves others and can strengthen social ties. These expanded networks can offer a range of support, from academic assistance to emotional and psychological support, thereby enhancing the overall social support system available to the student (Bavel et al., 2020; Mishra, 2020; Zhai and Du, 2022a). In a study by Mattanah et al. (2010), college students who actively sought help had higher levels of perceived social support, which in turn was linked to better academic and personal outcomes. Tinto argues that social integration is as important as academic integration for student retention and success (1993). Help-seeking behaviors contribute to this social integration by fostering connections and a sense of belonging within the academic community (Mattanah et al., 2010). These behaviors, therefore, not only address the immediate social, emotional and academic needs of students but also contribute to their long-term retention and success by building a robust support system.

Additionally, research has suggested that people with mental health issues are likely to endure social consequences, such as poor social support and negative social interactions (Kessler et al., 1998;

Eisenberg et al., 2007; Steger and Kashdan, 2009). Conversely, students with better mental health may be more inclined to engage in positive social interactions. These interactions may include academic collaborations and various facets of campus life (e.g., extracurricular activities, informal social gatherings). These findings suggest that mental health status plays a significant role in a student's ability to form and maintain supportive social relationships. According to Tinto (1993), social integration is essential to the success of a student's education journey. Improved mental health can facilitate this social integration by enabling students to more effectively connect with one another. This enhanced ability to connect, in turn, builds a stronger, more supportive community around the student.

Taken together, these protective factors (i.e., optimism, help-seeking behaviors, social support) may not only mediate the effects of mental health and risk perception on academic persistence but also intertwine with each other directly or indirectly, as outlined by previous research. Therefore, this current study integrated Tinto's theory of student departure and current literature to structure hypothesized paths between these protective factors as critical elements influencing students' decisions to persist with their academic pursuits amid the challenges of mental health conditions and the perceived risks of COVID-19. Accordingly, our research questions are as follows:

**Research Question 1.** Do students' mental health conditions correlate with risk perception of COVID-19?

**Research Question 2.** Do students' mental health conditions and risk perception of COVID-19 contribute to academic persistence?

**Research Question 3.** Does optimism mediate the effect of mental health conditions and risk perception of COVID-19 on help-seeking behaviors?

**Research Question 4.** Do protective factors (i.e., optimism, help-seeking behaviors and social support) mediate the effects of mental health conditions and risk perception on academic persistence?

## Methods

### Data collection and participants

The Office for Research Protections at the authors' institution approved this cross-sectional study. We assert that all procedures contributing to this work comply with the ethical standards of the relevant national and institutional committees on human experimentation and with the Helsinki Declaration of 1975, as revised in 2008. We used convenience sampling (e.g., university listserv, social media) to recruit participants nationwide from October 2020 to January 2021 before the spring semester, given that this fall cohort likely experienced more common and shared challenges during the early phase of the pandemic. After providing informed consent, participants completed an online survey via the Qualtrics platform. All participation is voluntary, and respondents could withdraw from the study at any time. On the informed consent form, participants were asked about their willingness to enter into a raffle for a chance to win one of five $10 Amazon gift cards. The inclusion criteria for this study stipulated that participants must be over the age of 18. They must be students who are currently enrolled at institutions of higher education regardless of students' demographics and institution types (e.g., public, private, national, regional, community, university, college). The current study sample comprised 1,051 college students from 45 states. All demographic characteristics are described in Table 1.

**Table 1.** Participant demographic information

| Characteristic | N | % |
|---|---|---|
| Race/ethnicity | | |
| American Indian or Alaska Native | 5 | 0.5 |
| Asian | 93 | 8.8 |
| Black or African American | 52 | 4.9 |
| Hispanic or Latino | 66 | 6.3 |
| Native Hawaiian or Pacific Islander | 1 | 0.1 |
| White | 752 | 71.6 |
| Multiracial | 52 | 4.9 |
| Additional race/ethnicity | 23 | 2.2 |
| Gender | | |
| Male | 197 | 18.7 |
| Female | 823 | 78.3 |
| Transgender | 12 | 1.1 |
| Additional gender | 9 | 0.9 |
| Prefer not to say | 3 | 0.3 |
| Sexual orientation | | |
| Heterosexual | 805 | 76.6 |
| Lesbian/gay | 35 | 3.3 |
| Bisexual | 118 | 11.2 |
| Queer | 34 | 3.2 |
| Additional sexual orientation | 27 | 2.6 |
| Prefer not to say | 25 | 2.4 |
| International student (hold an F-1 or J-1 visa) | | |
| Yes | 62 | 5.9 |
| No | 974 | 92.7 |
| First-generation college student | | |
| Yes | 271 | 25.8 |
| No | 758 | 72.1 |
| Not sure | 12 | 1.1 |
| Having a documented and diagnosed disability | | |
| Yes | 156 | 14.8 |
| No | 881 | 83.8 |
| Housing | | |
| On-campus residence hall/apartment | 101 | 9.6 |
| On/off-campus fraternity/sorority house | 5 | 0.5 |
| On/off-campus co-operative house | 4 | 0.4 |
| Off-campus apartment/house | 886 | 84.3 |
| Other (please specify): | 45 | 4.3 |
| Employment status | | |
| Yes | 674 | 64.1 |
| No | 366 | 34.8 |
| Current instruction mode experience[a] | | |
| In-person instruction | 410 | |

(Continued)

**Table 1.** (Continued)

| Characteristic | N | % |
|---|---|---|
| Remote synchronous instruction (i.e., interact in "real time" via videoconference) | 948 | |
| Remote asynchronous instruction (i.e., no scheduled meeting time) | 606 | |
| Other | 54 | |

*Note:* The total number of answers for each characteristic may be less than the resulting sample size (N = 1,051) due to participants' nonresponse.
[a]Given that participants were asked to check all that apply for current instruction mode experience, the percentages for this variable were not provided.

### Measures

A demographic questionnaire was utilized to collect information from participants regarding their age, race and ethnicity, sex and gender, disability status, housing and employment and learning environment (i.e., online/in-person/hybrid).

### Mental health conditions

The Mental Health Inventory-5 (MHI-5) is an abridged version of the 38-item MHI to assess mental health conditions (Berwick et al., 1991). The MHI-5 is a brief and effective measurement with the advantage of allowing a faster mental health assessment (Berwick et al., 1991), and it has been adopted widely in many settings and deemed valid and reliable for use in different subgroups (e.g., age, gender) and cultures (Rivera-Riquelme et al., 2019). The MHI-5 is a 5-item scale that measures psychological well-being and distress, such as anxiety, depression and emotional control, and it uses a 6-point Likert scale (1 = *all of the time* to 6 = *none of the time*). The validity of the MHI-5 is good and similar to the full 38-item MHI (Hartley, 2013). McDonald's omega ($\omega$) reliability of this scale was. 87 in the current sample.

### Risk perception of COVID-19

The Personal Risk Perception scale has been developed to gain information regarding personal-level perceived risk of infectious diseases (Tyler and Cook, 1984; Morton and Duck, 2001; Oh et al., 2015). This measurement was adapted in this current study to construct an index of perceived risk of COVID-19 specifically, using a 5-point Likert scale to assess the extent to which participants believe themselves to be personally at risk of COVID-19 infection (1 = *strongly disagree* to 5 = *strongly agree*): "(a) The problem of COVID-19 is serious to me; (b) I am worried that I would be affected by COVID-19; (c) It is likely that I would be affected by COVID-19; and (d) I have felt that COVID-19 is dangerous." McDonald's $\omega$ reliability of this scale was. 83 in the current sample.

### Academic persistence

The Student Integration Model Measurement (SIMM) is a 9-item scale that measures the degree of students' academic persistence at institutions of higher education (Cabrera et al., 1992). Based on the Institutional Integration Scale (Pascarella and Terenzini, 1980) and Tinto's theoretical framework of students' academic persistence (1975), the original 8-item SIMM was developed and validated by Cabrera et al. (1992) to measure five key academic persistence variables (i.e., one item for Academic Integration, two items for Social Integration, one item for Institutional Commitment, two items for Goal Commitment, one item for Intent to Persist) as core determinants of students' decisions to continue their education.

Cabrera et al. (1992) provide empirical backing for these academic persistence variables as these variables collectively offer a holistic view of the factors that influence a student's decision to remain at an institution, closely mirroring the multifaceted nature of retention. In the current study, the SIMM includes an additional item from the study of Pascarella and Terenzini (1980), "My non-classroom interactions with faculty have had a positive influence on my personal growth, values and attitudes" to recognize the essence of faculty interactions in shaping students' social integration experiences. This additional item had the highest factor loading within the corresponding scale, indicating its strong predictive capability (Pascarella and Terenzini, 1980). Adhering to the Institutional Integration Scale and the original SIMM, the current SIMM uses a 5-point Likert scale that ranges from 1, *strongly disagree*, to 5, *strongly agree.* The SIMM items ask questions regarding students' perceptions of academic development, faculty and peer relationships, institutional commitment, goal commitment and intent to persist in their institutions. McDonald's ω reliability of the SIMM was. 77 in the current sample.

### Perceived social support

The Multidimensional Scale of Perceived Social Support (MSPSS) is a 12-item measure of perceived adequacy of social support from three sources, including family, friends and significant other (Zimet et al., 1988). The MSPSS uses a 5-point Likert scale ranging from 1, *strongly disagree*, to 5, *strongly agree.* The MSPSS has three subscales reflecting support from family, friends and significant other. The MSPSS shows good internal consistency for the three subscales as well as for the scale as a whole. Zimet et al. (1988) reported good test–retest reliability over a 2- to 3-month interval (family: $\alpha = .85$, friends: $\alpha = .75$, significant other: $\alpha = .72$). McDonald's ω reliability of the MSPSS was. 88 in the current sample.

### Help-seeking behaviors

The General Help-Seeking Questionnaire (GHSQ) is an 11-item scale that measures help-seeking intention from both formal and informal help sources, in addition to miscellaneous help sources (Wilson et al., 2005). One of the helping sources (i.e., youth worker) was taken off from the questionnaire because the target population in this study are college students deemed as emerging adults. The GHSQ has been adopted to assess help-seeking behaviors in college student populations (Nguyen et al., 2019). The GHSQ has two subscales that measure help-seeking behaviors in terms of two problems (i.e., personal-emotional problems, suicidal ideation problems), and it uses a 7-point Likert scale that ranges from 1, *extremely unlikely*, to 7, *extremely likely.* In this current study, only the subscale (i.e., personal-emotional problems) was adopted as suicidal ideation was not examined. The survey of this study included six items (i.e., Partner, Friend, Parent/Family, Mental health professional, Would not seek help, Other sources). Wilson et al. (2005) reported satisfactory scale reliabilities: the scale as a whole (Cronbach's alpha = .85, test–retest reliability assessed over a 3-week period = .92), and the subscale (personal-emotional problems, Cronbach's alpha = .70, test–retest reliability assessed over a 3-week period = .86). According to Wilson et al. (2005), the GHSQ is found to have satisfactory validity (i.e., convergent, divergent, predictive and construct validity). McDonald's ω reliability of the GHSQ was. 62 in the current sample.

### Optimism

The Revised Life Orientation Test (LOT-R) is one of the most widely used measures of dispositional optimism (Scheier et al., 1994). The LOT-R consists of 10 items, among which three items assess optimism, three items assess pessimism, and four are filler items. It uses a 5-point Likert scale that ranges from 0, *strongly disagree*, to 4, *strongly agree.* Total scores ranged from 0 to 24; a higher score indicates a higher level of self-report optimism. Previous research suggested that the LOT-R is a valid and robust measurement that can be used to assess dispositional optimism among college students (Herzberg et al., 2006; Glaesmer et al., 2012). McDonald's ω reliability of the LOT-R was. 84 in the current sample.

### Sample size

The results of the A-priori sample size calculator for SEM (Soper, 2021) suggested that the recommended minimum sample size was 403 with a small effect size = .20, statistical power = .80, number of latent variables = 6, number of observed variables = 20, probability level = .05. Therefore, the current study sample size ($N = 1,051$) was sufficient to perform structural equation modeling (SEM) to test our hypotheses.

### Statistical analysis

In this current study, we followed Kline's SEM data analysis strategy (2016) to specify our hypothesized model. The hypothesized mediation model followed a two-step procedure involving confirmatory factor analysis (CFA) and SEM. In the measurement model, A six-factor CFA tested the proposed measurement model, including six latent variables and 20 observed variables: five for MHI-5, four items for the Personal Risk Perception scale, three for SIMM, two for LOT-R, three for GHSQ and three for MSPSS. All six latent variables' covariances were freely estimated to develop the measurement model of 20 observed variables. A full SEM was then applied to test the hypothesized model. A disturbance covariance for two exogenous latent variables (mental health conditions and risk perception of COVID-19) was freely estimated as there were unanalyzed associations assumed between these two exogenous variables (Kline, 2016). CFA and SEM were performed with SPSS Amos, version 26 (IBM). Figure 1 shows the hypothesized full structural equation model and standardized regression coefficients.

The parameter values were estimated through maximum likelihood methods. Several goodness-of-fit indices were used to assess how well the models described the input data set. The goodness of fit of the models was evaluated by using (a) a comparative fit index (CFI; values of. 90 or above suggest a good fit) (Bentler, 1990); (b) a root mean square error of approximation (RMSEA; values of. 08 or less suggest a good fit) (Steiger, 1990); and (c) the standardized root mean square residual (SRMR; values of. 80 or less suggest a good fit) (Martens, 2005). Normality checks were given in Amos to assess skewness, kurtosis and multivariate normality. A 2-sided $p < .05$ was considered statistically significant.

## Results

### Preliminary analyses

The result of Little's Missing Completely at Random (MCAR) test was not significant ($\chi^2 = 1,200.555$, df = 1,137, $p = .093$), suggesting missing data (0.19% within all responses) was MCAR (Tabachnick et al., 2007). Because of MCAR, we utilized the expectation–maximization (EM) algorithm (a more sophisticated imputation method) to impute missing data to obtain unbiased estimates of

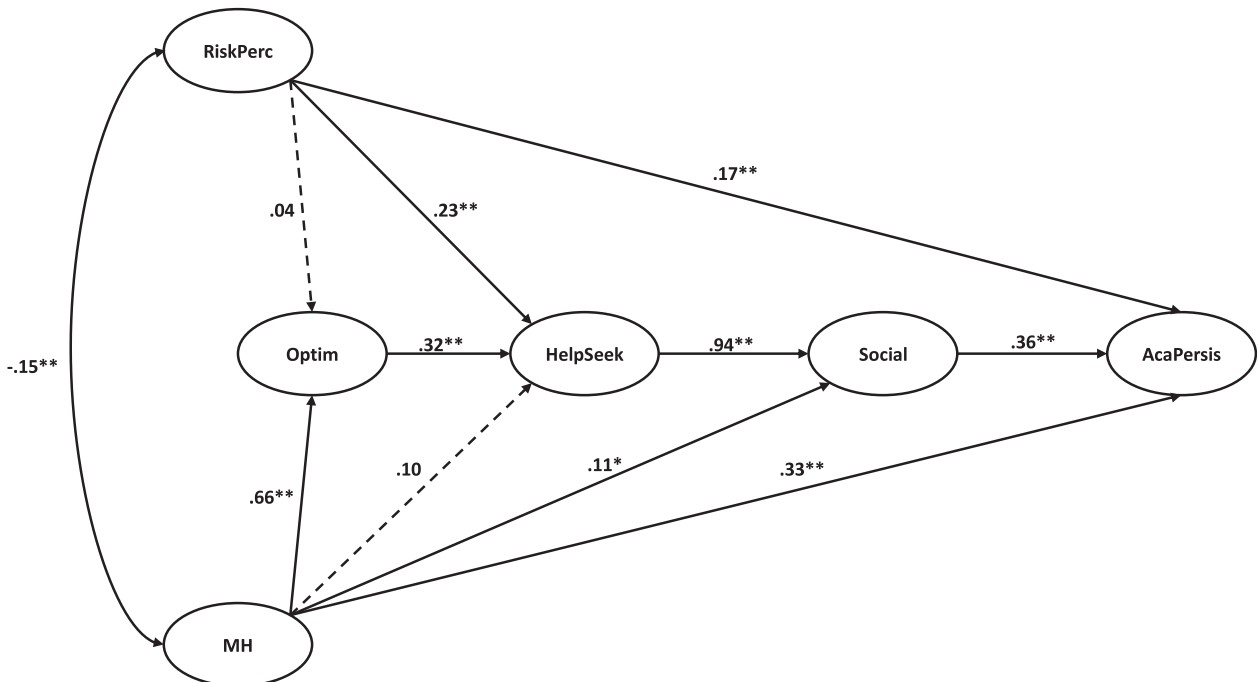

**Figure 1.** Hypothesized structural equation model with standardized regression coefficients.

*Note:* Observe variables (indicators) and error terms were omitted for simplicity of representation. RiskPerc, risk perception of COVID-19 measured by the Personal Risk Perception scale; MH, mental health conditions measured by MHI-5; Optim, optimism measured by LOT-R; HelpSeek, help-seeking behaviors measured by GHSQ; Social, social support measured by MSPSS; AcaPersis, academic persistence measured by SIMM. ∗$p$ < 0.01. ∗∗$p$ < 0.001.

parameters of interest (Tabachnick et al., 2007; Kline, 2016). Results from normality assessments showed that all variables' absolute value of skewness was less than three, ranging from 0.06 to 2.72, and that of kurtosis was less than 10, ranging from 0.25 to 7.67; thus, all assumptions were met, and further analyses could be conducted (Kline, 2016). Means, standard deviations and zero-order correlations for all variables are shown in Table 2.

### Structural equation modeling

CFA was performed to test the goodness of fit of the measurement model before estimating the hypothesized structural model. Six latent variables (i.e., mental health conditions, risk perception, optimism, help-seeking behaviors, social support, academic persistence) and 20 observed variables were included in CFA. The construct, mental health conditions, was estimated by five observed variables corresponding to five items of the MHI-5, respectively. Similarly, the construct, risk perception of COVID-

19, was estimated by four observed variables corresponding to four items of the Personal Risk Perception scale, respectively. Applying the item parceling technique, we estimated the construct, optimism, via two observed variables corresponding to two subscales of the LOT-R (i.e., optimism, pessimism), respectively. Likewise, the construct, social support, was estimated by three observed variables corresponding to three subscales of the MSPSS (i.e., family, friends, significant other), respectively. The construct, help-seeking behaviors, was estimated by three observed variables corresponding to two items and one subscale, respectively. These two items assessed professional help-seeking and non-help-seeking, and the subscale assessed non-professional help-seeking (i.e., partner, friend, parent/family). The construct, academic persistence, was estimated by three observed variables corresponding to one item and two subscales, respectively. This one item assessed students' intent to persist, and the subscales assessed commitment (i.e., goal and institutional commitment) and integration (i.e., academic and social integration). Final results suggested an overall good fit to

**Table 2.** Correlations and descriptive statistics

| Variable | M | SD | 1 | 2 | 3 | 4 | 5 | 6 |
|---|---|---|---|---|---|---|---|---|
| 1. Mental health conditions | 49.78 | 20.21 | 1 | | | | | |
| 2. Risk perception of COVID-19 | 23.54 | 4.28 | −0.199* | 1 | | | | |
| 3. Social support | 66.17 | 12.83 | 0.271* | 0.094* | 1 | | | |
| 4. Help-seeking behaviors | 26.46 | 5.55 | 0.157* | 0.192* | 0.593* | 1 | | |
| 5. Optimism | 13.41 | 4.83 | 0.530* | −0.06 | 0.329* | 0.221* | 1 | |
| 6. Academic persistence | 34.85 | 5.95 | 0.345* | 0.094* | 0.366* | 0.328* | 0.265* | 1 |

*$p$ < .01.

the data: $\chi^2$ (152) = 788.683, $p$ < .001, CFI = .923, SRMR = .063 and RMSEA = .063, 90% CI (.059,. 068). All 20 standardized factor loadings were statistically significant.

## Model fit

The overall model was evaluated with hypothesized pathways between latent variables to examine the structural portion of the model (Figure 1). This hypothesized overall evidenced a strong fit: $\chi^2$ (156) = 799.836, $p$ < .001, CFI = .922, SRMR = .063 and RMSEA = .063, 90% CI (.058,. 067). When compared with three alternative models, the hypothesized model was superior, as evidenced by better goodness-of-fit indices. In the first alternative model, there was no direct path from risk perception and mental health conditions to academic persistence. In the second and third alternative models, there was a direct path from risk perception and mental health conditions to academic persistence, respectively. Results confirmed that the hypothesized model was the best fitting.

## Direct effect

Results showed that risk perception of COVID-19 was negatively associated with mental health conditions. The direct effects of mental health conditions on academic persistence ($\beta$ = .33, $p$ < .001), optimism ($\beta$ = .66, $p$ < .001) and social support ($\beta$ = .11, $p$ < .01) were statistically significant. The direct effect of risk perception of COVID-19 on academic persistence ($\beta$ = .17, $p$ < .001) and help-seeking behavior ($\beta$ = .23, $p$ < .001) were statistically significant. The direct effect of mental health conditions on help-seeking behaviors ($\beta$ = .10, $p$ = .07) and of risk perception of COVID-19 on optimism ($\beta$ = .04, $p$ = .25) were, however, not significant. Further, optimism positively predicted help-seeking behavior ($\beta$ = .32, $p$ < .001) and help-seeking behavior positively predicted social support ($\beta$ = .94, $p$ < .001). Social support, in turn, positively predicted academic persistence ($\beta$ = .36, $p$ < .001). The model explained 36% of the variance in academic persistence ($R^2$ = .36), revealing a large effect size (Cohen, 1992).

## Mediation effect

We conducted bias-corrected bootstrapping to analyze the significance of mediation effects. In Amos, 2000 bootstrap data samples were generated by randomly sampling with replacement from the data set. Table 3 shows standardized parameter estimates of both the direct and indirect effects in the hypothesized model. The 95% CI for each indirect path indicated these indirect effects, except the one from risk perception of COVID-19 to help-seeking behaviors, were statistically significant. The direct effect of mental health conditions on academic persistence was significant, supporting a partial mediation effect. Namely, mental health conditions contributed to academic persistence directly and indirectly through heightened levels of protective factors (i.e., optimism, help-seeking behaviors and social support). The direct effect of mental health conditions on help-seeking behaviors was not statistically significant, but the indirect effect on help-seeking behaviors was significant through optimism, indicating full mediation; namely, mental health conditions contributed to help-seeking behaviors only indirectly through greater levels of optimism.

The direct effect of risk perception of COVID-19 on academic persistence was also significant, supporting a partial mediation effect. The direct effect of risk perception of COVID-19 on help-seeking behaviors was statistically significant, but the indirect effect on help-seeking behaviors was not significant through optimism. Additionally, results revealed the significant indirect effect of risk perception of COVID-19 on academic persistence only for these mediators together: help-seeking behaviors and social support ($\beta$ = .25, $p$ < .001), but not for these mediators together: optimism, help-seeking behaviors and social support ($\beta$ = .013, $p$ > .05).

## Discussion

This study is one of the first to investigate the relationships between students' mental health, protective factors and retention in responding to the risk of COVID-19 infections. Our main findings revealed that students' mental health conditions and risk

**Table 3.** Standardized direct and indirect effects of the model

| Path | | Direct effect | Indirect effect | Total effect |
|---|---|---|---|---|
| Optimism | ← Mental health conditions | 0.66** | | 0.66 |
| Help-seeking intention | ← Mental health conditions | 0.10 | 0.21* | 0.31 |
| Social support | ← Mental health conditions | 0.11* | 0.30** | 0.41 |
| Academic persistence | ← Mental health conditions | 0.33** | 0.15** | 0.48 |
| Optimism | ← Risk perception of COVID-19 | 0.04 | | 0.04 |
| Help-seeking intention | ← Risk perception of COVID-19 | 0.23** | 0.01 | 0.24 |
| Social support | ← Risk perception of COVID-19 | | 0.23* | 0.23 |
| Academic persistence | ← Risk perception of COVID-19 | 0.17** | 0.08** | 0.26 |
| Help-seeking intention | ← Optimism | 0.32** | | 0.32 |
| Social support | ← Optimism | | 0.30* | 0.30 |
| Academic persistence | ← Optimism | | 0.11** | 0.11 |
| Social support | ← Help-seeking intention | 0.94** | | 0.94 |
| Academic persistence | ← Help-seeking intention | | 0.34** | 0.34 |
| Academic persistence | ← Social support | 0.36** | | 0.36 |

*$p$ < .01.
**$p$ < .001.

perception contributed to academic persistence directly and indirectly through certain protective factors.

### Mental health and risk perception of COVID-19

The findings from our study highlighted a negative correlation between mental health conditions and risk perception of COVID-19 among college students. More specifically, an increase in the perceived risk of COVID-19 corresponded with a worsening of students' mental health. This suggests that those who perceive COVID-19 as more threatening and personally impactful are likely to experience a decline in emotional well-being, exemplified by symptoms such as depression and anxiety. Given that risk perception of this novel infectious disease has a substantial influence on mental health, our results align with previous studies indicating the severe psychological implications of various aspects of the pandemic on students (Cao et al., 2020; Zhai and Du, 2020). There is an urgent need for mental health care and services to address the consequences of the pandemic, while authoritative bodies such as the U.S. CDC and higher education institutions need to disseminate accurate and scientific knowledge to inform individuals' risk perception of COVID-19 while promoting mental health through targeted outreach programs. Such efforts are essential to control the spread of the virus effectively while ensuring the mental well-being of individuals.

### Mental health and academic persistence

The findings from our study indicated a positive relationship between students' mental health conditions and academic persistence. This suggests that students who maintain better mental health tend to demonstrate greater persistence in their academic pursuits. The COVID-19 pandemic has engendered considerable uncertainty concerning the duration of these transformed college experiences. The mental health of students, influenced by such uncertainty, can have a profound and long-lasting effect on their commitment to their universities and their studies (Hartley, 2013). As a consequence, students grappling with deteriorating mental health conditions are less likely to persist in their academic endeavors (Hartley, 2013). Therefore, these individuals are more prone to discontinuing their studies at their institutions during the pandemic (Tinto, 2017). In light of these findings, higher education institutions must devise strategic interventions to more effectively screen students in need of mental health support. By doing so, they can furnish these students with targeted aid and interventions, thereby amplifying student retention rates.

### Risk perception of COVID-19 and academic persistence

Interestingly, our study revealed that risk perception of COVID-19 positively and directly influenced academic persistence. This unexpected result suggests that students with a heightened perception of the risk associated with COVID-19 may be more likely to persist academically in higher education. One plausible explanation could be that certain emotions linked to risk perception, such as fear and concern, might ignite a motivation to adapt and persevere, thereby sparking proactivity among college students (Lebel, 2017). In spite of the convergence of challenges presented by the pandemic, students demonstrating increased proactivity might tenaciously continue their pursuit of academic goals and degree completion (Tinto, 2017). This can be attributed to the substantial investments made by students and their families in their education, coupled with a keen understanding of the crucial role that higher education plays in influencing future economic returns and opportunities for success (Abbiati and Barone, 2017).

### Optimism, help-seeking behavior and social support

Results from SEM and bootstrapping procedures revealed that optimism, help-seeking behaviors and social support partially mediate the relationship between students' mental health and academic persistence. These findings suggest that students struggling with poor mental health may be less inclined to seek the necessary support for academic perseverance. Notably, our study found that optimism fully mediates the relationship between students' mental health and help-seeking behaviors, suggesting that mental health conditions do not appear to directly influence help-seeking behaviors but rather indirectly through enhanced levels of students' optimism. The pandemic has instigated a range of repercussions, such as school closures, shifting instruction modes, isolation and loss, which exacerbate students' mental health issues (Cao et al., 2020; Zhai and Du, 2020). Pervasive pessimism and negative emotions can hinder help-seeking behaviors due to feelings of hopelessness, helplessness and perceived lack of agency (Wilson et al., 2005). These findings underline the importance of outreach programs targeting students with poor mental health and limited access to resources. These results may also inform therapeutic models employing skills such as mindfulness and positive psychology, fostering students' optimism. This enhanced optimism could help students recognize their potential, develop coping strategies and seek support in navigating the challenges presented by the altered college experience during the COVID-19 pandemic.

The results suggested a partial mediation effect of help-seeking behaviors and social support between COVID-19 risk perception and academic persistence. These individual protective factors appear crucial in students' academic persistence during the pandemic. A heightened perception of COVID-19 risk might spur help-seeking behaviors, fostering social support and positively impacting academic persistence (Tinto, 1993, 2017). Given the emotional challenges tied to the pandemic, encouraging help-seeking behaviors can fortify students' support systems, enhancing their resilience and promoting academic persistence (Bavel et al., 2020). These findings underscore the necessity of fostering such behaviors and social support to mitigate the pandemic's adverse effects on students' academic progress.

Our results indicated no significant association between COVID-19 risk perception and optimism; however, risk perception did positively predict help-seeking behaviors, with optimism failing to mediate this relationship. Despite studies suggesting optimism might reduce risk perception and health behaviors in certain individuals (Park et al., 2021), our findings propose that instilled optimism does not mediate the effect of perceived risk on help-seeking behaviors among college students. A feeling of worry can be adaptive and lead to protective behaviors such as seeking help (Liao et al., 2014). It is possible that students who are more optimistic might also have worries about the risk of infectious diseases, such as COVID-19, thus increasing help-seeking behaviors (Liao et al., 2014). With the potential of pandemic fatigue decreasing perceived risk and health behaviors like help-seeking (Petherick et al., 2021), higher education institutions and public health officials should proactively disseminate accurate and scientific COVID-19 information to encourage help-seeking behaviors (Crane et al., 2021).

## Limitations

This study has several limitations. First, while the cross-sectional nature of our data precludes definitive causality examination, our use of a structural mediation model, guided by established theory, provides valuable insights into potential causal paths between constructs. It should be noted that alternate path directions could also be plausible, underscoring the need for further research. Longitudinal data, in particular, would be advantageous for assessing temporal links and confirming the directionality of these associations. Second, the study employed scales developed and standardized during pre-COVID-19 times. While their application remains plausible during the COVID-19 era, interpretation of the findings must be undertaken with caution. Namely, the applicability of the SIMM (Cabrera et al., 1992), MHI-5 (Berwick et al., 1991), MSPSS (Zimet et al., 1988) and LOT-R (Scheier et al., 1994), among others, may vary with contemporary student populations, considering that some foundational studies date back several decades. The evolving educational environment requires caution when these measurements from the past are applied to current retention dynamics. Third, the use of the SIMM for measuring academic persistence as a proxy for student retention should be acknowledged as an imperfect indicator of retention. According to Cabrera et al. (1992), the SIMM explained 38% of the variance in student academic persistence, indicating that other contributing factors of academic persistence were not captured by this measurement. Finally, our findings' generalizability may be limited due to the predominance of self-identified white, female students in our participant pool. To enhance the findings' generalizability, future research would benefit from the inclusion of a more diverse participant pool, such as students of color and males.

## Conclusion

This current study sheds light on critical factors shaping academic persistence among college students during a global crisis, emphasizing the important role of students' mental health and protective factors in their academic persistence and retention. These findings underscore the urgency for higher education institutions to promote students' mental health and protective behaviors, which helps enhance students' academic persistence and support student retention. The development and implementation of targeted outreach programs and interventions are warranted to address these pressing issues around enrollment and educational inequalities, a pressing concern during the pandemic and in the face of potential future global crises.

**Open peer review.** To view the open peer review materials for this article, please visit http://doi.org/10.1017/gmh.2024.12.

**Data availability statement.** The data that support the findings of this study are available from the corresponding author upon reasonable request.

**Acknowledgments.** We thank Drs. Diandra Prescod, Julia Bryan, Mary Beth Oliver and Liza Conyers for their feedback on the early draft of this study.

**Author contribution.** Y.Z. conceived of the idea and conducted the literature review and data analysis. Both authors contributed to the final manuscript.

**Financial support.** None.

**Competing interest.** The authors declare no competing interest.

**Ethics statement.** The Office for Research Protections at the authors' institution approved this cross-sectional study. The authors assert that all procedures contributing to this work comply with the ethical standards of the relevant national and institutional committees on human experimentation and with the Helsinki Declaration of 1975, as revised in 2008.

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
