## [Reviewer Report]

10-July-2023

Prof Dr Judy Bass & Prof Dr Dixon Chibanda

Editors-in-Chief, Cambridge Prisms: Global Mental Health

Dear Prof Dr Bass and Prof Dr Chibanda,

We wish to submit a new research manuscript entitled “The role of mental health and protective factors in student retention during a global crisis” for consideration by Cambridge Prisms: Global Mental Health.

We confirm this work is original and has not been published elsewhere nor is it currently under consideration for publication elsewhere.

Given the global perturbations caused by the pandemic, the mental health implications on students and their corresponding effects on academic persistence have become significant points of concern. It is estimated that during the academic year 2020–2021, around 30% of U.S. college students confronted mental health problems, which has further amplified the challenges these students face, leading to a steep decline in enrollment rates.

In response to these pressing issues, our study aims to fill the existing research gap by examining the effects of students' mental health, risk perception of COVID-19, and protective factors on academic persistence—an indicator of student retention—using a sample of 1051 U.S. students from 45 states. The findings of this research underscore the vital role of mental health and protective factors in promoting student retention, and suggests that universities should consider developing ongoing targeted programs to address these needs during this global crisis and beyond.

Our study is particularly relevant to Cambridge Prisms: Global Mental Health, as it contributes to a deeper understanding of the interplay between mental health, protective factors, and academic persistence during a pandemic or similar crisis situations. The implications are of interest to educators, administrators, policymakers, and mental health professionals as they navigate the challenges of supporting students' mental health and retention during unprecedented times.

Furthermore, our research offers insights to a broader audience, such as families with college students, who can better support their children’s education during such times, and the students themselves, who can better understand the protective factors that bolster their resilience.

Thank you for considering our submission. We look forward to the potential opportunity to contribute to Cambridge Prisms: Global Mental Health.

Best regards,

Yusen Zhai, PhD, NCC, LPC

Assistant Professor, Department of Human Studies

Director, UAB Community Counseling Clinic

The University of Alabama at Birmingham

JoLynn V. Carney, PhD

Professor, Department of Educational Psychology, Counseling, and Special Education

The Pennsylvania State University

---

## [Reviewer Report]

Nov-16-2023

Dr. Judy Bass & Dr. Dixon Chibanda

Editors-in-Chief, Cambridge Prisms: Global Mental Health

Dear Dr. Bass and Dr. Chibanda,

Thank you for your editorial guidance and reviewers’ comments for our manuscript “The role of mental health and protective factors in student academic persistence and retention during a global crisis”. We greatly appreciate the opportunity to submit a revised version of this manuscript to Cambridge Prisms: Global Mental Health.

We are pleased to present to you this revised manuscript. The comments were very encouraging and reinforcing that this article would be a timely contribution to a deeper understanding of the potential role of mental health and protective factors in student academic persistence and retention during a time of public health crisis.

Major changes and additions to the revised manuscript:

We have:

1. provided additional details to increase clarity surrounding the framing of the study and instruments used to measure constructs in our model.

2. provided additional details in the Introduction and Current Study sections to justify the study outcome variable.

3. elaborated on the rationale of the specification of our hypothesized model grounded in Tinto’s theoretical framework and current literature.

We are particularly appreciative of the editorial and review team for the thoughtful and constructive comments and for the invitation to revise and resubmit our manuscript. Together, the feedback and suggestions have been instrumental in revising our manuscript, and all comments have been addressed.

Thank you for considering our revised manuscript. We look forward to the potential opportunity to contribute to Cambridge Prisms: Global Mental Health.

Best regards

Yusen

Yusen Zhai, PhD, NCC, LPC

Assistant Professor, Department of Human Studies

Director, UAB Community Counseling Clinic

The University of Alabama at Birmingham

JoLynn V. Carney, PhD

Professor, Department of Educational Psychology, Counseling, and Special Education

The Pennsylvania State University

---

## [Reviewer Report]

December-19-2023

Dr. Judy Bass & Dr. Dixon Chibanda

Editors-in-Chief, Cambridge Prisms: Global Mental Health

Dear Dr. Bass and Dr. Chibanda,

Thank you for your editorial guidance and reviewers’ comments for our manuscript “The role of mental health and protective factors in student academic persistence and retention during a global crisis”. We greatly appreciate the opportunity to submit a revised version of this manuscript to Cambridge Prisms: Global Mental Health.

We are pleased to present to you this revised manuscript. We are particularly appreciative of the editorial and review team for the thoughtful and constructive comments and for the invitation to revise and resubmit our manuscript. The comments were very encouraging and helped improve this manuscript further. We have attended to all the comments and incorporated all the suggestions into this revised manuscript.

Thank you for considering our revised manuscript. We look forward to the potential opportunity to contribute to Cambridge Prisms: Global Mental Health.

Best regards,

Yusen

Yusen Zhai, PhD, NCC, LPC

Assistant Professor, Department of Human Studies

Director, UAB Community Counseling Clinic

The University of Alabama at Birmingham

JoLynn V. Carney, PhD

Professor, Department of Educational Psychology, Counseling, and Special Education

The Pennsylvania State University